

# On the use of high frequency surface wave oceanographic research radars as bistatic single frequency oblique ionospheric sounders

Stephen R. Kaeppler[1], Ethan S. Miller[2], Daniel Cole[1,3], and Teresa Updyke[4]

[1]Department of Physics and Astronomy, Clemson University, Clemson, South Carolina, USA.
[2]Formerly at Johns Hopkins University Applied Physics Laboratory, now with Systems and Technology Research, Beavercreek, Ohio, USA.
[3]Now with U.S. Space Force
[4]Old Dominion University, Virginia, USA.

**Correspondence:** Stephen R. Kaeppler (skaeppl@clemson.edu)

**Abstract.** We demonstrate that bistatic reception of high frequency oceanographic radars can be used as single frequency oblique ionospheric sounders. We develop methods that are agnostic of the software defined radio system to estimate the group range from the bistatic observations. The group range observations are further used to estimate virtual height and equivalent vertical frequency at the midpoint of the oblique propagation path. Uncertainty estimates of the virtual height and equivalent vertical frequency are presented. We apply this analysis to observations collected from two experiments, run at two locations in different years, but utilizing similar software defined radio data collection systems. In the first experiment, 10 days of data were collected in March 2016 at a site located in Maryland, USA, while the second experiment collected 20 days of data in October 2020 at a site located in South Carolina, USA. In both experiments, three Coastal Oceanographic Dynamics and Applications Radars (CODARs) located along the North Carolina coast of the US were bistatically observed at 4.53718 MHz. The virtual height and equivalent virtual frequency were estimated in both experiments and compared with contemporaneous observations from a vertical incident Digisonde ionosonde at Wallops Island, VA, USA. We find good agreement in both experiments between the virtual height derived from the oblique CODAR observations versus the virtual height observed with the Digisonde at the same frequency. Variations in the virtual height from CODAR observations and the Digisonde are found to be nearly in phase with each other. We conclude from this investigation that observations that oceanographic radar can be used as single frequency oblique incidence sounders. We discuss applications with respect to investigations of traveling ionospheric disturbances, studies of day-to-day ionospheric variability, and using these observations in data assimilation.

## 1 Introduction

Understanding the spatial and temporal variations of the ionospheric electron density remains an ongoing challenge in the space weather community. It has been known since the 1950s that disturbances propagate through the ionospheric electron density; these structures have been termed traveling ionospheric disturbances (TIDs) (e.g., Munro, 1950). It has been established that there are two dominant spatial and temporal scales for TIDs: medium scale TIDs (MSTIDs) and large-scale TIDs (LSTIDs) (Hunsucker, 1982; Hocke and Schlegel, 1996; Harris et al., 2012; Frissell et al., 2014; Otsuka, 2021). However, understanding





the physical processes responsible for the generation of these disturbances and the sources of the disturbances remain open questions. Some observations have also suggested non-propagating disturbances in the electron density (e.g., Harris et al.,
25  2012).

Making progress toward addressing these questions has been frustrated by the relatively sparse observations. In the last 10 years, vertical incidence ionosondes have been used in networks of various spatial sizes (Cervera and Harris, 2014; Reinisch et al., 2018; Belehaki, Anna et al., 2020). Incoherent scatter radars provide height resolved observations, but these facilities are constrained to a single observational location and small field of view (Kirchengast et al., 1995; Kirchengast et al., 1996;
Nicolls et al., 2004; Nicolls and Heinselman, 2007; Vlasov et al., 2011). SuperDARN has been used to examine TIDs over larger spatial scales, which has produced climatological results particularly for MSTIDs (e.g., Bristow et al., 1994; Frissell et al., 2014). GPS TEC has high spatial density (e.g., Crowley et al., 2016; Coster et al., 2017; Figueiredo et al., 2017), but tends to be biased toward the electron density associated with the F-region peak and the topside of the ionosphere (Nickisch et al., 2016; Belehaki, Anna et al., 2020); this technique can be relatively insensitive to the bottomside of the ionosphere.

High frequency (HF) radiowave propagation experiments have been developed to understand the electron density of the bottomside ionosphere. One of the earliest techniques measured the Doppler shift of a carrier frequency of timing signals (e.g., WWV) that had high phase coherence (Georges, 1968; Crowley and Rodrigues, 2012). Using the Doppler shift, it is possible to derive the ionospheric velocity (Davies, 1990). Chilcote et al. (2015) used a similar technique to measure Doppler shifts of clear channel AM radio stations in the northeast sector of the US to derive properties of TIDs. TIDs have also been investigated
using the frequency and angular sounding (FAS) technique in which observations of angular deflections, the Doppler shift frequency, and the group delay changes are related back to an oscillating mirror disturbance moving through the ionosphere (Beley et al., 1995; Galushko et al., 2003; Paznukhov et al., 2012; Huang et al., 2016). This technique has been primarily used in association with radio telescope observatories (Beley et al., 1995; Obenberger et al., 2019). Recently, the TechTIDE effort have also been undertaken to use networks of ionosondes to study TIDs, with both vertical incidence and oblique sounding
(e.g., Reinisch et al., 2018; Belehaki, Anna et al., 2020).

Over the past two decades, advances in receiver technologies and computer storage have evolved to the point that inexpensive, direct-sampling digital software defined radios are able to sample and store the full 30 MHz HF spectrum. Simultaneously, inexpensive satellite-based precision navigation and timing devices have simplified and reduced the cost of synchronizing stations to produce bistatic observations. These techniques have enabled the development of low-cost, low-power ionosondes, at
single or multiple frequencies (Vierinen, 2012; Hysell et al., 2016; Saito et al., 2018; Bostan et al., 2019; Chartier et al., 2020). For example, one such experiment that has ultized software defined radios is a network of bistatic HF beacon transmitters and receivers located in Peru, that have been used to investigate conditions associated with the development of equatorial spread-F (Hysell et al., 2016, 2018, 2021).

One transmission that can be received bistatically is from coastal HF radar stations that are used for ocean wave and current
diagnostics within several hundred kilometers of the coast line (e.g.,  Barrick, 1972; Barrick et al., 1977; Gurgel et al., 1999). These systems operate on the principle of Bragg scatter from ocean waves and operate at a variety of frequencies between 4–50 MHz, with some of the most useful frequencies for ionospheric diagnostics operating between 4–5 MHz. A typical radar signal





at 4–5 MHz has a bandwidth of approximately 25 kHz, which is similar to the current generation of ionosondes (Reinisch, 2021), and a waveform repetition frequency (WRF) of 1 Hz. The modulation is a linear frequency modulated continuous wave

(FMCW) chirp, resulting in a one-way range resolution of $\sim$ 10-12 km, thus making these transmission suitable as a single frequency oblique ionosonde. Although ionospheric sounding has been reported on internet blog posts (RFSPACE, 2011; Estevez, 2017), to our knowledge, there have not been any investigations in the peer-reviewed literature that has evaluated the efficacy of these HF oceanographic radars as potential vertical or oblique single frequency ionospheric sounders.

The purpose of this investigation is to test whether bistatic observations of coastal HF radars can be used as high time

resolution single frequency oblique ionospheric sounders. We present observations of the group range and polarization splitting of the skywave mode at two locations using similar software defined radio systems that observed three Coastal Oceanographic Dynamics and Applications Radar (CODARs) located along the east coast of the US. Using the group range and the known location of the transmitters, we quantify the virtual height and effective vertical frequency using the so-called secant law (Davies, 1990). We compare the CODAR derived virtual heights and frequency with similar vertical incident HF soundings

from the Wallops Island DPS256 Digisonde to validate our methodology. We conclude by discussing some of the observed features and their implications for improving understanding of spatial and temporal variations of the electron density in the ionosphere.

## 2   Methodology

### 2.1   Basic Signal Processing

A classic "chirped" (linear frequency modulated) continuous wave (CW) radar operates on the principle of a continuously varying transmitter of known frequency characteristic, for example,

$$\tilde{s}_{tx}(t) = s_0 e^{-j2\pi f(t)t} \tag{1}$$

where $f(t) = \kappa t$ is a linearly varying frequency. For a monostatic (shared transmit and receive antenna with gain $G$) radar illuminating $i = \{1, N\}$ targets with radar cross sections ($\sigma_i$) at ranges $R_i = c_0/2t_i$, the received signal is,

$$\tilde{s}_{rx}(t) = \frac{s_0 G^2}{4\pi} \sum_i^N \frac{\sigma_i}{R_i^4} e^{-j2\pi\kappa(t-t_i)t} \tag{2}$$

Mixing (multiplying) the transmitted signal with the conjugate of the received signal together yields a series of discrete fre-

quency components at baseband corresponding to the ranges to the individual targets,

$$\tilde{s}_{bb}(t) = \frac{s_0^2 G^2}{4\pi} \sum_i^N \frac{\sigma_i}{R_i^4} e^{j2\pi\kappa(t_i)t} \tag{3}$$

The range can then be calculated from,

$$R_i = \frac{\kappa c_0}{2f(t_i)} \tag{4}$$





This may be generalized to the case of the ionospheric propagation channel between two (bistatic, in the radar lexicon) stations, the "targets" represent individual distinct propagation modes supported by the channel between the two stations. The FMCW radar technique has a number of advantages over pulsed radars, including the high average power on the target. The Doppler shift is due to a change in phase path to the target,

$$\Delta f = \frac{1}{f}\frac{dR_\phi}{dt}. \tag{5}$$

Practically speaking, the change in the phase path may be calculated by comparing (fitting) the observed phase on subsequent coherently-processed pairs (sequences) of returned pulses (waveforms). The WRF sets the Doppler shift aliasing frequency, which is half the WRF.

For the present analysis, a replica of the transmitted waveform is prepared and correlated with the received signal in the frequency domain in the following manner. The replica of a single waveform (pulse) is created, extending Equation 1 to include a raised-cosine window ($w(t)$) in order to reduce artifacts when Fourier transforming a function with finite support:

$$\tilde{\mathbf{s}}_{tx}[n] = s_0 w[n] e^{-j2\pi\kappa n^2}, \tag{6}$$

where $\kappa$ is the frequency sweep rate and $s_0$ is the magnitude and can be taken as unity. Using discrete sampling, $n$ is the sample index within the waveform.

In order to increase signal-to-noise ratio and perform Doppler shift calculations, a number ($M$) of sequential waveforms are integrated coherently. If each waveform contains $N$ samples, this is accomplished in the frequency domain thusly,

The replica is Fourier transformed and replicated into an $N \times M$ matrix:

$$\tilde{\mathbf{S}}_{tx} = \begin{bmatrix} \mathcal{F}\{\tilde{\mathbf{s}}_{tx}\} \\ \mathcal{F}\{\tilde{\mathbf{s}}_{tx}\} \\ \vdots \\ \mathcal{F}\{\tilde{\mathbf{s}}_{tx}\} \end{bmatrix} \tag{7}$$

Likewise, the entire coherent processing interval (CPI, $\tilde{\mathbf{S}}_{rx}$) is also (Discrete) Fourier transformed (in two dimensions) and the Hadamard (elementwise, ∘) product taken with the replica ($\tilde{\mathbf{S}}_{tx}$) to complete the convolution in the frequency domain.

$$\tilde{\mathbf{Q}} = \mathcal{F}\left\{\tilde{\mathbf{S}}_{rx}\right\} \circ \tilde{\mathbf{S}}_{tx} \tag{8}$$

The notation breaks down a bit when we go back to the time domain, where only the Fourier transform in the "fast time" (samples within a waveform) dimension is taken:

$$\tilde{\mathbf{R}} = \mathcal{F}^{-1}\left\{\tilde{\mathbf{Q}}\right\} \tag{9}$$

The result, $\tilde{\mathbf{R}}$, is an $N \times M$ matrix that contains the magnitude and phase versus range ($N$ elements) and Doppler ($M$ elements). Standard range-time-intensity (RTI) presentations of the data may be achieved by identifying or fitting the peak power at each range for each processing interval.



## 2.2 Virtual Height Estimation

To determine the virtual height, we make the simple approximation that the ionosphere is effectively a reflecting mirror at an altitude of $h$, which is defined as the virtual height (Davies, 1990). We determine the virtual height, $h$, using the following trigonometric relation (Davies, 1990),

$$\left(\frac{P}{2}\right)^2 = \sqrt{h^2 + \left(\frac{D}{2}\right)^2} \tag{10}$$

where $P$ is the group range, $P = c\Delta t$, with $\Delta t$ being the time delay of the ionospheric channel, and $D$ is the distance between the CODAR transmitter and the receive site. Considering that we obtain the group range, $P$, and we know the distance $D$

between the transmitter and receiver, this mirror model is the most straight forward model which is consistent with the observables. A more sophisticated treatment of this inversion problem using ray tracing is possible, but our approach is to consider the simplest model first. We assume that curvature of the Earth is not significant for this application considering that the ground range are typically less than 600 km (Davies, 1990), but we do use the Haversine formula to calculate the distance, $D$.

We also use the so-called secant law to calculate the equivalent vertical frequency for the oblique propagation path (Davies,

115 1990),

$$f_o = f_v \sec(\phi) \tag{11}$$

where $f_o$ is the oblique frequency, i.e., the frequency observed at the receiver, and $f_v$ corresponds to the vertical frequency of the ionospheric layer at the midpoint of the path between the transmitter and the receiver. The angle $\phi$ is derived as $\sin\phi = D/P$, where we note that $D$ is known and $P$ is obtained by radar signal processing, as described above in Section 2.1.

We use the E-region or surface wave as a means by which to calibrate for an absolute group range for the F-region prop-

agation mode. Each CODAR has a unique time delay set relative to GPS pulse-per-second (PPS), but those may not be well known ahead of time. The daytime E-region propagation mode is suitable since it occurs at a relatively stable and constant group range. To predict a group range for the E-region, $P_E^p$, we solve equation 10 using the known distance $D$ and an assumed virtual height, $h_E$, of 125 km. We can measure the group range of the E-region hop, $P_E^m$, and the calibration time delay, $\Delta$, can be determined as,

$$\Delta = P_E^m - P_E^p \tag{12}$$

Our results are relatively insensitive to the choice of $h_E$. This calibration factor can now be used to obtain $P_F = P_F^m - \Delta$, where $P_F^m$ is the measured group range for the F-region. Given $P_F$, the virtual height for the F-region can be estimated using equation 10.

The uncertainty of the virtual height, $\Delta h$, can be determined by propagating the uncertainty through equation 10,

$$\Delta P^2 = \left(\frac{4h}{P}\right)^2 \Delta h^2 + \left(\frac{D}{P}\right)^2 \Delta D^2 \tag{13}$$

where $\Delta h$ is the uncertainty of the virtual height, $\Delta P$ is the uncertainty in the group range, and $\Delta D$ is the uncertainty in

the ground range distance. For this investigation, we assume that $\Delta D$ is effectively zero, because we know the location of




the transmitters and the receiver locations to high levels of certainty. The uncertainty of the group range, $\Delta P$, corresponds to range uncertainty produced by dechirping the waveform. Since the bandwidth of the waveform is $\sim$25.733913 KHz, the corresponding group range uncertainty is $\Delta P \sim 11.7$ km (for one-way propagation). The propagated uncertainty for $f_v$ is,

$$\Delta f_v = \frac{f_o D^2 \Delta P}{P^2 \sqrt{P^2 - D^2}} \tag{14}$$

where $D$ and $P$ are known or observed, respectively.

## 2.3 Polarization

We also calculate Stokes parameters when we had observations with both loop antennas to determine whether the incoming wave has right-hand or left-hand circular polarization, corresponding to X- and O-mode propagation (in the northern hemisphere), respectively. For the convention of increasing phase, the loop antennas define an orthogonal coordinate system,

$$V = -2Im(V_x V_y^*) \tag{15}$$

where $V_x$ and $V_y$ corresponds to the complex voltage for the nominal north-south and east-west aligned antennas, respectively. Left-hand circular polarization (O-mode) corresponds to $+V$ and right-hand circular polarization (X-mode) correspond to $-V$.

## 2.4 Group Range Trace Extraction

After performing the radar signal processing, we obtain a time series of the pseudo group range, in which the time correction has not yet been performed, as discussed above. From a given transmitter, there can be two polarization modes per ionospheric channel. A Python-based "clicker" program was developed and implemented to extract the pseudo group range time series for each ionospheric propagation mode and polarization. Although more sophisticated methods could be used, including cluster algorithms or machine learning techniques, we find that this problem is well-suited to a more manual approach, similar to handscaling ionograms (Dandenault et al., 2020).

## 3 Experiments

The signal processing outlined in Section 2 is agnostic to the software defined radio data acquisition system; therefore, a wide variety of experiments can be envisioned and conducted. Table 1 describes the cost break down of the software defined radio system used at the Clemson Atmospheric Research Laboratory (hereafter referred to as "CARL") site, including the receive antennas, but excluding the cost of a host computer. Both systems, at both receive locations as discussed below, are based on a commercially-available Ettus Research Universal Software Radio Peripheral (USRP) model N210 which provides complex baseband samples over a simple Ethernet interface and a well-defined API. The standard build of the FPGA in the USRP N210 allows IQ sampling at a minimum of 250 kHz, which is then filtered and decimated in real time by a factor of 8 in software on a personal computer and recorded to disk. These raw, decimated, IQ samples are processed off-line using the algorithm



**Table 1.** Receive System at CARL (March 2022)

| Part Name | Cost |
|---|---|
| Ettus USRP N210 | $2,892 USD |
| Ettus GPSDO Kit (Jackson Labs FireFly-1a) | $1,280 USD |
| DX Engineering RF-PRO-1B Loop Antenna x 2 units | $1100 USD |
| Cabling, Misc Items | $500 USD |
| **Total:** | **$5772 USD** |

described previously. Other software defined radios could provide similar performance to the Ettus USRP system, but such an investigation is outside the scope of this paper.

The USRP N210 may be synchronized to GPS PPS and a stable 10-MHz timebase either via external inputs or an internal
option GPS Disciplined Oscillator (GPSDO) board (Jackson Labs FireFly-1a). Recordings start on the rise of a 1PPS pulse and continue for the duration of the coherent processing interval, usually 5–20 seconds. Besides bistatic time synchronization, the GPSDO also provides clock stability, which is essential for performing the signal processing described in Section 2.1. The oscillator native in the USRP N210 has a stability of 2.5 ppm which is insufficient for this application because the dechirped waveform will have large range resolution, driven by the jitter of the oscillator. The USRP is designed to be a component in
a receiving system and so requires a front-end low-noise amplifier to increase gain and reduce noise figure, even at HF. A suitable amplifier has about 20 dB of gain and a moderately high ($>$ 20 dBm) power at 1 dB compression.

We present results collected at two locations in different years, but using similar data collection systems. A quiet, exurban receiving site (hereafter referred to as "MSR") was established near the Johns Hopkins University Applied Physics Laboratory at approximately 39.34 N, 77.06 W. A short, commercially-available "active whip" (electric field probe) antenna about 2-
meters tall was installed at ground-level. The receiver was tuned to 4.53718 MHz, where three CODAR radars along the North Carolina coast share the same frequency allocation by a time-division multiple access (TDMA) scheme, with each radar's chirp start time delayed some arbitrary number of milliseconds from GPS PPS. These CODAR transmitters have the callsign: DUCK, LISL, and CORE. The second location was at the Clemson Atmospheric Research Laboratory (CARL). Two crossed loop antennas were used at this location, and the 4.53718 MHz CODAR band was also monitored.

Figure 1 shows a map detailing the locations of CARL and MSR relative to the CODAR transmitters and a diagnostic ionosonde, i.e., a DPS256 Digisonde, at the NASA Wallops Flight Facilty (hereafter referred to by its callsign "WP937"). The midpoints of the ionospheric channel from MSR to the CODARs are indicated by black Xs. The midpoints for the CODAR-CARL path form a north-south line located in North Carolina. The location of the receiver, transmitters, and diagnostic instrument are summarized in Table 2.



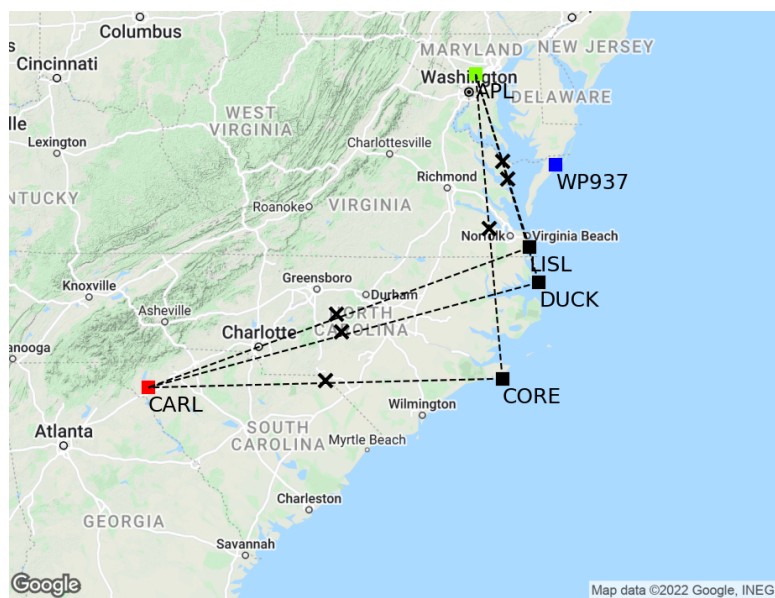

**Figure 1.** A map showing the physical locations of the CODAR transmitters (DUCK, CORE, and LISL), the Wallops Island Digisonde (WP937), and the receive locations of MSR and CARL. Midpoint locations along the great circle path are shown as black Xs.

**Table 2.** Receiver and Transmitter Sites

| Site Name | Function | Geographic Latitude | Geographic Longitude | Distance to MSR | Distance to CARL |
|-----------|----------|---------------------|----------------------|-----------------|------------------|
| MSR | Receive | 39.34 N | 77.06 W | – | – |
| CARL | Receive | 34.62 N | 82.83 W | – | – |
| DUCK | Transmit | 36.18 N | 75.75 W | 370 km | 664 km |
| LISL | Transmit | 36.69 N | 75.92 W | 311 km | 665 km |
| CORE | Transmit | 34.76 N | 76.41 W | 513 km | 587 km |
| WP937 | Diagnostic | 37.94 N | 75.47 W | 208 km | 755 km |

## 4 Results

The results presented correspond to 10 days of continuous data collected at the MSR site during 10-19 March 2016 and 20 days of continuous data collected at the CARL site for 05-24 October 2020. Results are shown from two locations to demonstrate that similar data collection systems produce consistent features. A representative example from 10 March 2016 at MSR and 08 October 2020 at CARL are presented in Figure 2 and Figure 3, respectively. These figures correspond to pseudo group range-time-intensity (RTI) on the y-x-z axis, respectively. The circuits are arranged in the following order from smallest GPS PPS delay to the largest: LISL, DUCK, and CORE. This arrangement is based on an internal time offset from GPS PPS, which has been provided by the Radio Operators Working Group.



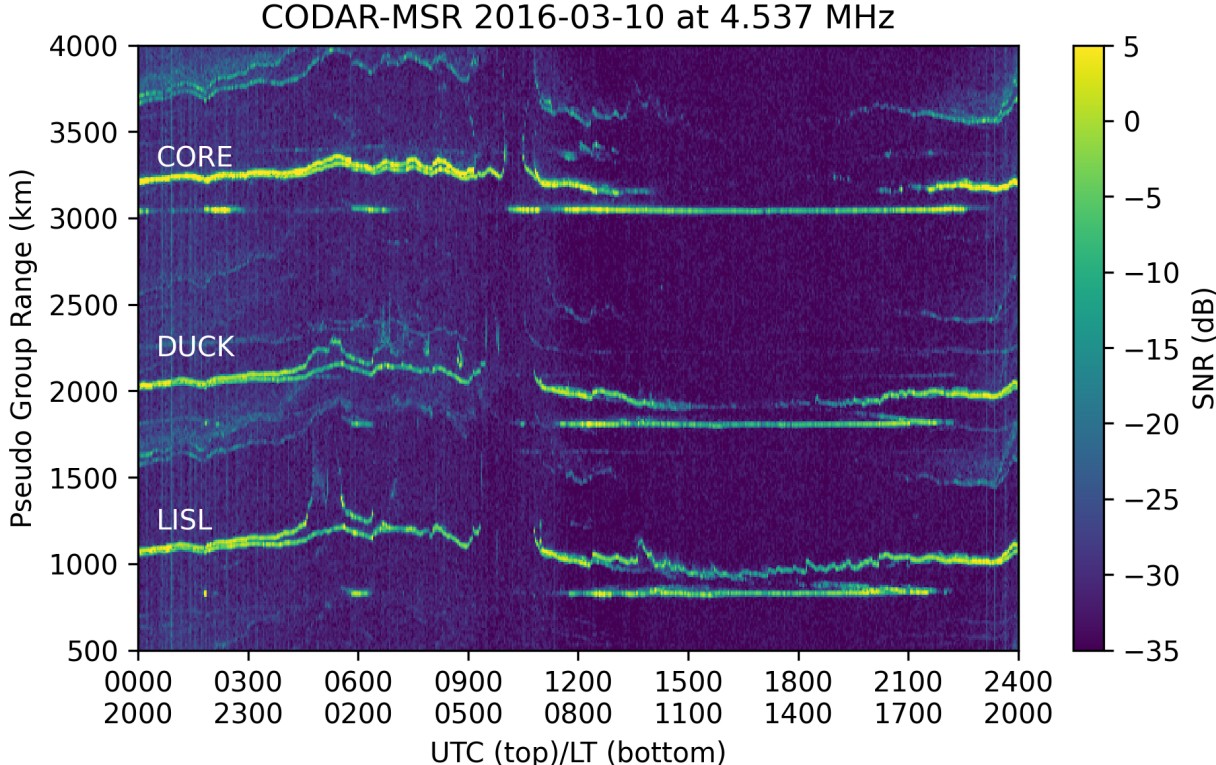

**Figure 2.** A 24-hour (psuedo) range-time-intensity (RTI) plot from three North Carolina ocean wave radar transmitters received at MSR on 4.53718 MHz on 10 March 2016. The pseudo group ranges corresponding to the LISL, DUCK, and CORE CODAR transmitters are listed. The figures are plotted with respect to UTC, but Eastern Time (LT) is also shown for reference. See the text for further discussion.

A few features are quite clear in Figure 2. Universal Time (UTC) is Local Time + 4 hours; so, the time series begins in the early evening local time, a little after sunset. We note that even though a single antenna was used, mode splitting becomes apparent in the pseudo group range, as indicated by two distinct propagation modes at the same time. Throughout the night, 0000–1100 UTC, two stable $F$-region propagation modes are present, separating slowly in range as the bottom side of the $F$-region erodes due to recombination. Around 0430 UTC on the LISL circuit, the ordinary (O) mode begins to penetrate through the ionosphere before returning at 0530 UTC for another hour. After about 0630 UTC, only the extraordinary (X) mode closes the circuit until it disappears at 0930 UTC. On the DUCK circuit, the O-mode departs from the X-mode by 100–200 km during 0430–0530 UTC and after 0630 UTC, but does not penetrate the ionosphere as in the LISL case. This is due to the slightly longer great circle distance to DUCK, and attendant increase in the subionospheric zenith angle. Likewise, the O mode remains subionospheric until 0930 UTC on the CORE circuit.

Also during the night of 10 March 2016, several sporadic-$E$ layers appear, around 0200–0300 UTC and 0530–0700 UTC. For the sake of discussion, assume that the geographic extent of the sporadic-$E$ layers covers the reflection points of all three





circuits (in this case, a north-south extent of about 70 km), we observe the appearance and dissipation at different probing

critical frequencies. By solving equation 11 this expression for $f_v$ with $f_{\mathrm{MUF}} = 4.53718$ MHz yields the "cutoff frequency"

for each circuit in the $E$-region. That is, equation 11 predicts the ionospheric critical frequency at which the fixed-frequency

circuit no longer completes. Since each circuit has a different great circle distance (and therefore subionospheric zenith angle),

it creates a sieve to investigate the rate of change of electron density.

The top row of Figure 3 shows a similar RTI as Figure 2 using the data collection system at the CARL. We find many

similar features between the observations from MSR and the observations from CARL. For example, a stable daytime E-region

due to solar production becomes clear between 1030–2230 UTC at pseudo group ranges of 1500 km, 2300 km, and 3600

km, respectively. D-region absorption is also evident during the day, as represented by the fading of the $F_2$ path or opening

of the propagation circuit during the daytime. Dawn and dusk show rapid changes in the pseudo group range which occur at

approximately 1030 UTC and near 0100 UTC, respectively. These rapid changes correspond to an increase and decrease of

photoionization caused by the sun rising and setting locally, respectively.

The bottom row of Figure 3 presents the polarization calculated from the cross-loop antenna configuration at CARL using

equation 15. The polarization provides additional insight into the propagation modes that were observed with the system at

CARL. We find at night, between 0000–1030 UTC and after 0100 UTC, that X-mode is the dominant propagation mode. For

this day, we also find some intervals where both O- and X-mode propagation were present including between 0700-1000 UTC.

During this interval in particular, the ionosphere may have been supporting a high/low ray configuration (Davies, 1990) in the

O-mode propagation which corresponds to the "hoop" feature in the pseudo group range. However, further investigation is

required to verify this hypothesis, which is outside of the scope of the current investigation.

### 4.1   Comparison with Wallops Island Digisonde

Figure 4 presents a comparison of 5 days of CODAR derived virtual height relative to virtual height observations provided

by the Wallops Island Digisonde, WP937. The five days chosen correspond to 15–19 October 2020. For each CODAR-CARL

circuit, the corresponding virtual height was calculated using equation 10 and the equivalent vertical frequency, $f_v$, was calcu-

lated using equation 11 above. We interpret the location of the virtual height and equivalent vertical frequency to correspond to

the midpoint of the great circle path between the CODAR transmitter and the CARL, as shown as black Xs in Figure 1. From

the Digisonde data, we extract the virtual heights for the frequency nearest to the equivalent vertical frequency derived from the

CODAR observations. We plot the power from this frequency slice as a function of time, similar to the height-time-intensity

figures discussed in Altadill et al. (2019). For the Digisonde data, we consider both polarizations, with O- and X-mode corre-

sponding to red and green, respectively, but gray out received power that is $< 30$ dB below the peak power. The black and blue

lines correspond to the virtual height for the X- and O-mode CODAR-CARL paths, respectively. This Digisonde processing

creates an equivalent single frequency time series which is shown in the top three rows for LISL, DUCK, and CORE, respec-

tively. The fourth row shows the equivalent vertical frequency derived for the X-mode for LISL, DUCK, and CORE as red,

blue, and green, respectively; the fifth row shows the equivalent vertical frequency derived for the O-mode.





**Figure 3.** Top Row: 24-hour (psuedo) range-time-intensity plot from three North Carolina CODAR transmitters received at CARL at 4.53718 MHz on 08 October 2020. Bottom Row: Polarization separation of O- and X-mode propagation as red and green, respectively. See the text for further discussion.

We find good agreement between the oblique CODAR derived virtual height and virtual height from the WP937 Digisonde. The consistency between the CODAR observations and the WP937 observations is best during the night, between roughly 0100 – 1030 UTC. We observe a gap in the CODAR observations during the day from 1030–2230 UTC, which is due primarily to absorption associated with the daytime D-region ionosphere. The propagation channel between the CODAR transmitter and





the CARL is open. Additional gaps in data sometimes correspond to the ionosphere not supporting propagation; for example, between ~0000–0600 UTC on 16 October 2020, the original RTIs show an open propagation channel during the night.

Figure 5 corresponds to 5 days of observations from MSR between 10–14 March 2016, looking at the same CODAR transmitters at 4.53718 MHz and presented in a similar format to Figure 4. In this case, because a single vertical antenna was used we present only a single virtual height, although there was propagation mode splitting evident in the pseudo group range, as shown above. We find good agreement when comparing the virtual height from WP937 with the equivalent virtual height from the CODAR observations. The night time sector, 0000–0900 UTC, is well resolved with these observation and there are fairly consistent features day-to-day for this 5 day interval. We also find that the daytime F-region ionosphere can be resolved, in particular for the LISL circuit between 1200–1800 UTC. For the case of MSR, LISL has the shortest circuit length; therefore, we expect the equivalent vertical frequency to be the largest. The equivalent vertical frequencies, presented in the bottom row, also show frequencies nearest to transmitter frequency, as indicated by the secant law, equation 11 above. With this analysis, spatial diversity of transmitters relative to receivers also provides frequency diversity.

The virtual height separation between the X- and O-mode is less pronounced for the CODAR observations from the CARL site versus the Digisonde observations or the from the MSR site. This can be partially attributed to oblique propagation. The propagation delay time (group range) will tend to increase because the ray will slow down as it nears the reflection point. However, as the ray paths become more oblique, the ray will tend to reflect at lower altitude thus resulting a similar group range between the X- and O-mode propagation.

For completeness, we include observations from CARL for October 5–9, October 10–14, October 20–24 as Supplemental Figures S01, S02, and S03, respectively. Observations from MSR are included as March 15–19 as Supplemental Figure S04.

## 5   Discussion

Our results have demonstrated that coastal HF surface wave radars, with appropriate waveforms, can be used as bistatic oblique ionospheric sounders at a single frequency. The RTIs shown in Section 4 are qualitative consistent with Figure 2 in Hysell et al. (2016), which is a HF beacon experiment that uses a psuedorandom code with 10 $\mu$s baud length, and Figure 5 from Bostan et al. (2019). Our analysis demonstrates that a time series of the virtual height can be obtained through bistatic reception of CODAR transmissions and our observations produce similar virtual height as the virtual height extracted from the WP937 Digisonde observations at a single frequency.

### 5.1   Applications

Observations such as these can be used to understand the spatial scale sizes of disturbances that affect the bottomside electron density. These disturbances may include propagating disturbances, i.e., TIDs, or so called non-traveling disturbances which can have scale sizes less than 500 km (Harris et al., 2012). The spatial distribution of north-south CODAR-receiver midpoints can enable investigations to test whether these disturbances are propagating. We would expect to see a time delay between the Digisonde observations of virtual height relative to the CODAR observations of virtual height, or between virtual height



observations derived solely from multiple CODAR observations. Although it is important to note that while CODAR observa-
tions in the 4 MHz band have a minimum sweep period of 1 second, a full Digisonde sweep is of the order of 5 minutes. If,
hypothetically, the CODAR midpoint relative to a Digisonde were separated by ∼200 km (∼ 2° latitude), for a LSTID, with a
nominal propagation speed of 500 m/s purely north-south (e.g., Figueiredo et al., 2017), the time delay would be ∼ 6 minutes.
This time interval is resolvable by CODARs alone with ≤ 1 minute sampling, but approximately 1-2 datapoints of digisonde
observations would cover the passage of a wavefront of the LSTID, at a fixed frequency due to the ∼5 minute revisit time.

Figure 6 presents an example of an assumed TIDs observed on 06 October 2020 between 0000–1200 UTC, during local night.
We find perturbations in the virtual height derived using CODAR data are in phase with virtual height observations derived
from the WP937 digisonde, for the same frequency. Figure 6 shows the CODAR derived virtual height tracks the WP937 X-
mode time series very well. Previous ionosonde observations have shown that TIDs cause perturbations of isodensity contours
that are in phase as the ionosonde frequency increases (e.g., Reinisch et al., 2018; Altadill et al., 2019; Belehaki, Anna et al.,
2020). Our observations are consistent with TechTIDE height-time-intensity (see Figure 6 Altadill et al., 2019).

The implications of a small time delay may suggest a preferential propagation direction, i.e., a crest of the TID may be over
both the Wallops Digisonde WP937 and the midpoints of the CODAR at the same time. Another possible explanation is that the
ionosphere was moving upward and downward over a spatially extended region that covers the CODAR midpoints and WP937.
Either of these hypotheses can be investigate further with additional CODAR observations and ionosonde observations, but this
investigation is outside of the scope of this paper. Additional data sources, such as GPS TEC or airglow imagers, may be able
to provide added insight into these structures. The observations from HF coastal radars could be used as an additional signal
source for TechTIDE or other similar efforts to understand and mitigate effects from TIDs.

Nearly continuous bistatic observations can be used to quantify the day-to-day variability of the bottomside ionosphere. TIDs
are one of the most significant sources of day-to-day variability (e.g., Harris et al., 2012; Frissell et al., 2014; Reinisch et al.,
2018). A recent investigation by Zawdie et al. (2020) examined bottomside day-to-day variability using the SAMI-3/WACCM
coupled ionosphere-thermosphere model and comparing the model output with ionosonde observations. They concluded that
one of the best parameters to quantify day-to-day variability are virtual height measurements at a fixed frequency, which this
investigation has demonstrated can be produced from bistatic reception of the CODAR transmissions. Qualitatively, Zawdie
et al. (2020) suggested there was more variability during the local night time sector, which is consistent with the observations
in this investigation. CODAR derived virtual heights can be used to assess day-to-day variability, although limited to a narrow
range of frequencies, but with higher time resolution and more diverse spatial coverage.

A third application is use of these data in assimilative ionosphere models of the bottomside ionosphere. A number of recent
investigations have suggested methodologies to assimilate data collected from HF beacons to inform regional ionosphere
models (Nickisch et al., 2016; Fridman et al., 2016; Mitchell et al., 2017; Hysell et al., 2016, 2018; Munton et al., 2019;
Hysell et al., 2021). Our simple analysis illustrates the potential efficacy of high bandwidth FMCW transmitters as a possible
data source for these models. Additionally these observations provide a suitable source to examine other propagation models,
including the effects of TIDs (Huang et al., 2016; Zawdie et al., 2016; Psiaki, 2019).





## 5.2 Limitations and Advantages

One of the key limitations of passive reception of single frequency sounding is limited probing of the ionospheric plasma.
Clearly, a swept frequency vertical incidence sounder is able to probe the bottomside altitude distribution of the ionospheric electron density, while this method is limited to a smaller range of electron densities. From the secant law, equation 11 above, the equivalent vertical electron density that can be probed cannot be greater than the transmit frequency of the CODAR. For this case, this limits our observations to 4.5 MHz for the case of a nearly vertical incident sounding. As the distance between the transmitter and receiver increases the equivalent vertical frequency will also decrease. One way to circumvent this issue
would be to use additional frequencies, as has been done in the investigation by Chartier et al. (2020).

A few other limitations include degraded signal performance during the day time due to D-region absorption. Second, the unambigous Doppler resolution is approximately 0.5 Hz for a 1 second FMCW signal. Pulsed-Doppler signal processing techniques can be applied (Richards, 2005); this step is left for a future investigation.

While there are some important limitations with this technique, there are also some advantages that make use of these sig-
315 nals appealing. First, our results demonstrate an additional use of an existing network of coastal surface wave HF transmitters; therefore cost for a transmitter is not required. Second, the overall cost of the system is relatively inexpensive and with advances in software defined radio technology, particularly with respect to clock stability, it is likely that the system cost will continue to go down over time. In fact, it is possible that the current generation software defined radio dongles with 0.5 ppm TXCO may be used for this application. Third, the spatial distribution of stations can be configured to more optimal probe the ionospheric
isodensity. For the first order analysis we have performed, the controlling factor of the equivalent vertical electron density is the distance from the CODAR transmitter to the receiver. Fourth, the data we obtain has high temporal resolution. In principle, we could obtain a data point at 1 second cadence, which is significantly faster than a typical frequency sweep of an ionosonde corresponding to 5-15 minutes depending on the sweep rate. Finally, presented in this investigation only shows use of polarization and group range data. Other state parameters can be obtained, including angle of arrival and Doppler shift, which can be
used in a more rigorous estimation of the ionospheric electron density as has been performed in other investigations (Galushko et al., 2003; Paznukhov et al., 2012; Huang et al., 2016; Hysell et al., 2018, 2021).

## 6 Summary and Conclusions

We present an investigation demonstrating that bistatic observations of existing high frequency coastal oceanographic radars, with suitable waveform characteristics, can be used as single frequency oblique ionospheric sounders. The techniques we
describe in this investigation are agnostic of the type of software defined radio system used, although good frequency stability is required. We present techniques for extracting the group range from the bistatic observations and using the E-region as a means to provide an absolute time delay for the F-region propagation mode. The virtual height and equivalent vertical frequency at the midpoint of the oblique path are also estimated using the group delay observations and the known location of the transmitters and receivers. Uncertainty estimates of the virtual height and equivalent vertical frequency are also derived.



We performed an experiment in which we collected 10 days of data in March 2016 from a site in Maryland, USA (MSR) and 20 days of data collected in October 2020 from a site near Clemson, South Carolina, USA (CARL). For both experiments, we used a similar hardware setup utilizing an Ettus USRP N210 software defined radio, including the GPSDO unit. We obtained bistatic observations of Coastal Ocean Dynamics Applications Radars (CODARs). Our observations for both intervals focused on one frequency band at 4.53718 MHz which included three CODAR transmitters located on the Coast of North Carolina, USA, with callsigns: DUCK, CORE, and LISL. The Digisonde located at Wallops Island, VA (WP937) was used as the diagnostic to compare and validate with the observations collected from oblique CODAR-MSR(CARL) propagation channels.

For the analysis, we estimated the virtual height and equivalent vertical frequency from the CODAR-MSR(CARL) path. For the Digisonde data, we extracted the virtual height at the frequency nearest to the CODAR derived equivalent vertical frequency and produced a time series of the virtual height of the Digisonde data. Upward or downward virtual height changes observed by the Digisonde were also observed by the CODAR derived observations with small time delays. The agreement was best during the night, which was partially attributed to significant D-region absorption during the day, thus causing no suitable CODAR-CARL path. Our results show that disturbances in the virtual height appear to be correlated over a spatial scale length of ∼350 km.

To our knowledge, this is one of the first investigations that has compared and validated bistatic HF observations from oceanographic radars with ionosonde measurements. The application of this investigation may be useful for expanding spatial coverage for traveling ionospheric disturbance studies, day-to-day variability studies, or within data-assimilation routines. Additionally, HF coastal radars may be used by the scientific community or radio amateur, i.e., HamSci, as a suitable RF source for ionospheric sounding.

*Data availability.* Processed and analyzed CODAR observations from CARL and MSR that were used in this investigation can be found at: https://doi.org/10.5281/zenodo.6341875. The Wallops Island Digisonde Data for October 2020 can be found at: https://data.ngdc.noaa.gov/ instruments/remote-sensing/active/profilers-sounders/ionosonde/mids12/WP937/individual/ and for March 2016: https://data.ngdc.noaa.gov/ instruments/remote-sensing/active/profilers-sounders/ionosonde/mids08/WP937/individual/. We wish to thank Dr. Terrance Bullett for writing the dgsraw which was used to process the Digisonde data: https://data.ngdc.noaa.gov/instruments/remote-sensing/active/profilers-sounders/ ionosonde/software/Digisonde/dgsraw-0.5.1.tar.gz. We wish to thank Lowell Digisonde for collecting ionosonde data as part of the Global Ionospheric Radio Observatory (GIRO) (Reinisch and Galkin, 2011).

*Author contributions.* SRK collected and analyzed data from CARL and MSR, produced figures, and drafted the manuscript. ESM provided CODAR observations from the MSR site and drafted portions of the manuscript. DC developed the Python pseudo range extraction algorithm and applied to the CARL data set. TU provided timing information for the CODARs located along the East Coast of the US.

*Competing interests.* There are no competing interests.



*Acknowledgements.* S.R. Kaeppler was supported by Office of Naval Research Grant N00014-21-1-2546 to Clemson University. D. Cole was supported by an Undergraduate Student Research Award provided by the NASA South Carolina Space Grant Consortium. SRK wishes to acknowledge G. Bust for useful contributions to this manuscript. E.S. Miller was supported by Air Force Office of Scientific Research grant FA9550-14-1-0278 on Office of Naval Research Grant N00014-17-1-2123 to JHU/APL. E. S. Miller acknowledges fruitful discussions with V. Mecca and G. Ginet of MIT Lincoln Laboratory.





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





**Figure 4.** A virtual height-time derived using the oblique CODAR-CARL observations versus the WP397 Digisonde at Wallops Island, VA between 15–19 October 2020. In the top three rows, the X- and O-mode virtual height time series are plotted as black and blue lines, respectively. The WP937 X-mode and O-mode at the frequency nearest to the equivalent vertical frequency derived using the CODAR observations is shown as green and red, respectively. From top to bottom, row 1–3 shows the virtual height calculated from the LISL, DUCK,





**Figure 5.** A comparison of the virtual heights derived using the CODAR observations from MSR for 10–14 March 2016 and the virtual height from the WP937 Digisonde at Wallops Island, VA. The figure is in the same format as Figure 4, except there is not distinction between X- and O-mode. See the text for more details.

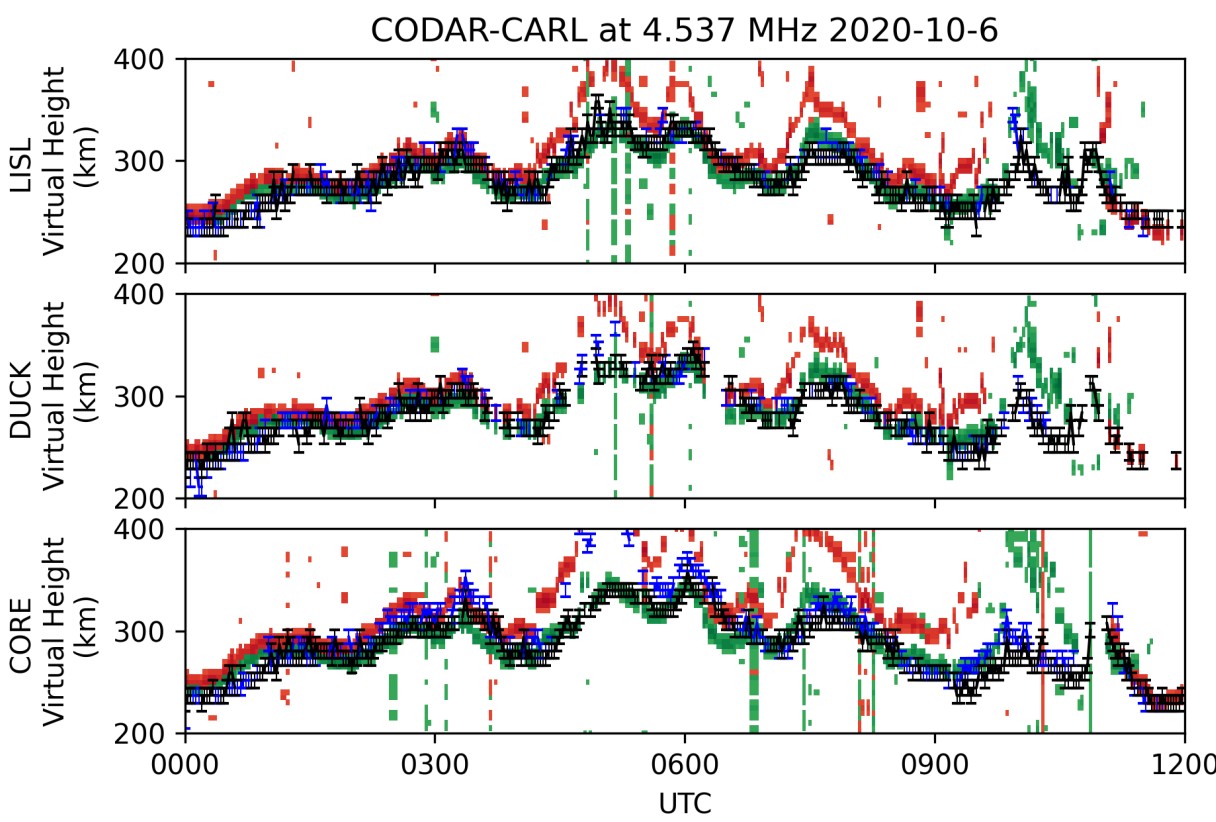

**Figure 6.** An example of a TID observed on 06 October 2020 from 0000–1200 UTC. The format is similar to Figure 4. The virtual height derived from CODAR observations for X- and O-mode are shown as black and blue lines, respectively. We also show uncertainty estimates for the virtual height. More details can be found in the text.