# Peer review of "On the use of high frequency surface wave oceanographic research radars as bistatic single frequency oblique ionospheric sounders"

_EGUsphere, 2022_

## Author Response (AR1)

**We wish to thank the reviewer for the excellent comments which we believe will help to make this manuscript stronger. Please find our responses as bolded.**

**Review 1 Comments:**

The paper describes the use of coastal ocean dynamics applications radar (CODAR) transmissions for ionospheric sounding. This is feasible because these radars operate at HF frequencies and the same signals that scatter from ocean waves can also be reflected from ionospheric plasma. The main point of the study is that there exist relatively many such transmitters and it is relatively inexpensive to use these transmissions for studies of the spatial and temporal structure of traveling ionospheric disturbances. The paper makes use of well known formulas for estimating vertical equivalent plasma frequencies and virtual heights from oblique propagation paths (secant law, Breit-Tuve theorem, and Martyn's theorem) that can be found e.g., in the textbook by Davies that is used as a reference.

The paper is well written and does not contain any significant flaws. I believe that your interpretation starting from line 215 is correct. The "hoops" are the two different propagation paths that are possible. The merging of these hoops should also allow you to estimate the peak O-mode cutoff. Similar behaviour is also seen in the E-region trace. My only criticism is that if the structure of TIDs is a sufficiently compelling science case, wouldn't you want to setup a network of fast ionosonde transmitters and receivers to optimally study them? I know this is a bit of an unfair point to make, as the point of the paper is to demonstrate a technique. In addition to the TechTIDEs references, I recommend looking into Andrew Heitmann's thesis and references therein for a review of recent work with multi-static oblique HF radio wave propagation used for studies of TIDs: https://digital.library.adelaide.edu.au/dspace/handle/2440/130401

My recommendation is accept as is.

PS. I just spent two hours writing a review in the textbox of the on-line review system. When I submitted the review, my text dissapeared as my session had expired. This is the second time it has happened to me. Please fix the system. My second review (that I had to rewrite from scratch) was significantly shorter than the first.

**Response to the Reviewer:**
**We wish to thank the reviewer for their review of this manuscript. We have added in a reference to the thesis by Heitmann near line 45 of the revised draft. Regarding your question of setting up a fast set of sounders, we certainly agree this is a better way to approach investigating TIDs. However, the purpose of this investigation is to focus on what we can derive from an existing set of HF transmitters, which has a fixed frequency**

**and waveform characteristics, which we do not control. We have added a few sentences throughout the manuscript to make the point that we do not control the transmit frequency or waveform characteristics. See lines 69, 315, and 340 of the revised manuscript. We are also sorry to hear you had trouble uploading your response, but that is a system that the EGU journal will need to update.**

**Reviewer 2 Comments:**

This is a very interesting paper that is of value to the ionospheric sensing community. It demonstrates that a low-cost software defined receiving system can be used to receive transmissions from other HF users, and contribute meaningful propagation path and travelling ionospheric disturbances (TIDs) information to the community. The paper is well written and structured.

Unfortunately I have only had the opportunity to read the paper once (my bad), but would like to read it at a later date, and may make further comments later.

I have four reservations that I'd like the authors to address:

There is a no information provided about how the signal transmitted by the CODAR propagates into the ionosphere. CODARs are designed to transmit signals at low elevation towards the sea so the transmitted waveform "adheres" to the sea surface, rather than into the ionosphere which can potentially introduce "clutter" signals that impact the interpretation of the sea scatter the radar seeks to exploit. One presumes the majority of the signal received is transmitted through the sidelobes (or backlobes) of the CODAR transmit array? Figures 2 and 3 (particularly the latter after 2200 UTC) reveals range spreading indicative of multi-path, likely from horizontally transmitted signal that is then backscattered off sea-waves into the ionosphere. It would be worth doing a simplified signal analysis that includes a) the transmit antenna pattern and b) backscatter coefficients to give a clearer picture of the transmission path.

**We have added a few sentences clarifying that we assume that the F-region time series, which we use in the analysis, comes predominantly from direct ionospheric propagation through a sidelobe or backlobe of the CODAR transmission antenna. We also added a sentence to guide the reader toward the multihop propagation and spreading, along with the spreading being potentially due to sea scatter, ground clutter, or possibly the presence of mid latitude spread-F. For midlatitude spread F, we would expect that to appear on the 1-F propagation path too though, so it seems less likely this is the culprit of the range spreading. Please see the paragraph around lines 215 of the revised manuscript.**

We also performed a very simple analysis to determine whether the scatter we are seeing could be associated with sea clutter or ground clutter. Simplistic, we might expect the ground clutter to occur at a group range of ~2 relative to 1-F mode. For the LISL link near 2200 UT, we find the 1E pseudo group range to be ~1500 km, 1F to be ~1650 km, and spread region to be ~1950 km (by eye). Using our analysis, the calibration factor is 790 km. Using this we find the calibrated group range to be 860 km and 1160 km for 1F and the spread region respectively. Calculating the virtual height we get 272 km and 475 km for the 1F and spread region, respectively. Dividing the spread region/1F, we get 1.74, so note quite the factor of two we expect, but close.

To test the sea scatter hypothesis, assume the scatter region is 50 km away from the transmitter. If we assume the virtual height of the reflection point is approximately the same, we can predict what we expect the group range should be for the sea scatter. WE find that

$$P\_SS^2 = P\_1F^2 + D\_CODAR^2 - D\_SS^2$$

Where SS corresponds to sea scatter. P_1F = 860 km, D_CODAR = 665, D_SS = 715 km (665+50), we get a predicted P_SS to be 900 km, and if we add back in our calibration factor in reverse, we get 1690, which is not near 1950 km, but much closer to the 1F, as we would expect since we are slightly extending the base of the triangle.

We find more consistency with the path being a multihop propagation path vs. sea clutter. However, considering that this mode is not always present and does produce spreading, it would be an interesting investigation to determine whether the ocean illuminated by the CODARs was more disturbed or not. A positive correlation may demonstrate that this is in fact sea scatter, but that should be reserved for a future investigation.

I would like to see more effort expended in interpreting the ionospheric processes responsible for some of the features observed in Figures 2 and 3. I realize the intent of the paper is to demonstrate the sensor capability, but having made reasonable efforts to describe some of the ionospheric processes it is a shame that this has not been taken to a logical conclusion. For instance, there is no reference to the multi-hops observed. The "hoops" are a well known phenomenon where the "nose" of the F2 low "breathes" in and out such that the maximum usable frequency "oscillates" about the transmission frequency. Unfortunately I'm unable to locate a reference for this at present, but I'm sure there must be one available if the authors are prepared to put in the effort. I also note that the aforementioned range spreading on the F2-low mode around 2200 UTC looks like mid-latitude spread-F. Having said that, spread-F occurs typically after midnight local time. As discussed above, this spreading may be indicative of

multipath, which may indicate a weakness of the proposed receiving system in that it may be unable to unambiguously verify existence of spread-F, a phenomena often thought to be associated with TID's.

**We agree with the reviewer's perspective that some of the investigations should have been (will be) taken to completion. However, the purpose of the investigation is to describe a technique that can be applied to an existing set of HF transmitters used for a different purpose than ionospheric science. In fact, had we attempted what the reviewer suggested it would have effectively resulted in another paper or two. We decided to keep the focus on the technique in this paper. There are other ionospheric science investigations which can use these datasets, but those investigations should be conducted as stand-alone studies. The comments regarding the interpretation of spread-F can be found in the response to the reviewer above (see response #1) and the paragraph near line 215.**

4. Instead of operating the receivers as fixed frequency sounders, the authors could operate them as swept frequency sounders which may potentially yield improved ionospheric information. Have the authors considered this? What are the pros and cons for such operation?

**We have added a few sentences to clarify that we do not have control over the waveform characteristics or frequency of the CODAR transmitters. See lines 69, 315, and 340 of the revised manuscript. The waveform characteristics of the transmitter is published online (http://hfrnet.ucsd.edu/sitediag/stationList.php) and has been discussed in blog posts that were cited within the paper. One of the advantages of using these transmitters is that they already exist for a different purpose; we are merely using their signal as an illuminator of opportunity. Operating fast swept frequency sounders would certainly produce improved ionospheric information. The limitation of the transmitter frequency was also discussed in Section 5.2.**

1. I find the authors reference to "open" and "closed" propagation channels misleading. They appear to use "open" to mean there is no propagation path available. This contradicts the terminology used in the HF communications were "open" means "available for use". I suggest the authors use less ambiguous terms to clarify whether a propagation channel is available.

**We have updated the terminology in the paper to reflect that "open" means "available for use" and "closed" means "not available for use." Please see lines 245-250, near line 205.**

Some minor issues and grammatical comments:

the comment about GPS TEC on line 32 slightly misses the mark. GPS TEC is a path-integrated quantity that is strongly *influenced* by the peak electron density, rather than *biased* by it. The use of the word *bias* suggests that the estimated value is incorrect. **Fixed.**

Line 119 states "We use the E-region or surface wave as a means by which to calibrate for an absolute group range". Only the DUCK CODAR in Figure 2 shows any sign of (weak) surface wave signal (at 1600 km between 10 and 15 UTC) so I have doubts as to how useful the surface wave is for range calibration. The surface wave signal may be more useful at higher frequencies where there is less groundwave attenuation

**We changed this to "sea surface" wave since Ethan Miller has shown me data where you can clearly see the surface wave over the ocean at a constant range for data collected in Puerto Rico.**

Line 258: "qualitative" should be "qualitatively" – **Fixed**

Line 259: suggest replace "which is a" with "which illustrates the results from a". **Fixed**

Line 260: suggest replace "produce similar virtual height as the virtual height extracted" with "produce similar virtual height as that extracted" **Fixed**

Line 318: replace "go down" with decrease. **Fixed**

Line 319: replace "optimal" by "optimally". **Fixed**

Figure 4: please fix the caption, which the caption runs of the edge of the page.

**We hope this issue will be taken care of during another stage of the publication process.**

---

## Author Response (AR2)

We once again wish to thank the Reviewer for taking the time to review our manuscript. We appreciate the comments they have provided, particularly some of the wordsmithing comments. The Reviewer's comments are bold and our response is in normal font.

We want to make the editor aware that many of the figures in the revised manuscript have changed in color content, so these figures are colorblind friendly. We used the following website: https://colorbrewer2.org/#type=qualitative&scheme=Dark2&n=3 We have edited the text accordingly, which is noted in the track changes document.

**I am happy with the revisions made by the authors in response to my first review, and with their rationale for not making some of the revisions suggested. As such, I am happy to accept the paper subject to the minor revision suggestions as follows:**

**- It is worth pointing out that Figure 2 appears to show a descending sporadic-E layer between 1800 and 2200 for the DUCK and LISL site.**

Added: We also note that between 1800--2200 UTC the DUCK and LISL link show evidence of a descending layer.

**- Line 237 - "grayed out". This doesn't appear to be the case - the low SNR data appears to be excluded rather than grayed out.**

Changed "grayed out" to "excluded."

**- Line 37: suggest authors revise this to say "We find good agreement between the oblique CODAR derived and WP937 Digisonde virtual heights".**

Changed.

**- Line 285 and Figure 6 caption: possibly pedantic, but Figure 6 does not show an example of an assumed TID - rather, it shows a virtual height time plot that illustrates an assumed TID.**

Revised the sentence in the caption: "An example of virtual height observations of an assumed TID on 06 October 2020 from 0000--1200 UTC." And in the following paragraph: "Figure \ref{fig:CARL06Oct} presents an example of virtual height observations of an assumed TIDs observed on 06 October 2020 between 0000--1200 UTC, during local night."

**- Line 294, investigate should be investigated.**

Changed

**- Lines 316 and 325: suggest replace "while this method" with "while the single frequency sounding"**

Changed

**- Line 240 separate "even" and "though"**

Changed

**- Line 443, Heitmann thesis, please add "University of Adelaide" before "School of...."**

Added

**- Figure 5 caption - add "for the CODAR observations" after "O-mode".**
Added

**- Figure 6 caption - add "the top three panels of" before "Figure 4".**

Added